# Effects of Mindfulness-Based Cognitive Therapy on Major Depressive Disorder with Multiple Episodes: A Systematic Review and Meta-Analysis

**DOI:** 10.3390/ijerph20021555

**Published:** 2023-01-14

**Authors:** Hui-Wen Tseng, Fan-Hao Chou, Ching-Hsiu Chen, Yu-Ping Chang

**Affiliations:** 1School of Nursing, Fooyin University, Kaohsiung 831301, Taiwan; 2College of Nursing, Kaohsiung Medical University, Kaohsiung 807378, Taiwan; 3School of Nursing, University at Buffalo, The State University of New York, Buffalo, NY 14260, USA

**Keywords:** mindfulness-based cognitive therapy, major depressive disorder, systematic reviews, meta-analyses

## Abstract

This study synthesizes the effect of mindfulness-based cognitive therapy (MBCT) on depression and suicidal ideation among patients with major depressive disorder (MDD). During treatment, patients with MDD may experience repeated episodes, negative counseling, and suicidal ideation, which can lead to further depression and ultimately affect quality of life. Recent studies have shown that MBCT can improve the level of depression and suicidal ideation in patients with MDD. A systematic review and meta-analysis of randomized controlled trials was conducted. The literature search for articles up to December 2021 was performed in the following electronic databases: Airiti Library, PsycINFO, CINAHL, Cochrane Library, PubMed/MEDLINE, ProQuest, and the Index of the Taiwan Periodical Literature System. Records were independently evaluated by two reviewers. Disagreements were resolved through consensus. The quality of study was evaluated using the Modified Jadad Scale score. A meta-analysis was performed using Review Manager Version 5.3.5 software with a random-effects model. Thirteen studies (1159 participants) investigating MBCT for patients with MDD were included. The MBCT sessions lasted 1.5–2.5 h and were delivered by therapists five times per week for 8 weeks. The meta-effects of MBCT among patients with MDD showed significant improvement in depression and suicidal ideation. Future research should evaluate the long-term effects of MBCT. MBCT is relatively convenient and effective for preventing and alleviating depression and suicidal ideation. Further research can provide detailed suggestions for effective MBCT implementation.

## 1. Introduction

### 1.1. Impact of Major Depressive Disorder on Mental Health

According to the World Health Organization (WHO) in 2020, major depressive disorder (MDD) affects over 264 million people and is currently the leading cause of disability worldwide [1]. More than 60% of individuals who have attempted suicide globally struggle with MDD [2]. Depression can occur as a single episode, recurrent, or chronic disease. It was estimated that by 2030, MDD will be the most common cause of disability worldwide. A World Health Assembly (WHA) resolution passed in May 2013 called for a comprehensive, coordinated response to mental disorders at the country level.

The global burden of MDD due to functional impairment, particularly loss of productivity, as well as symptom severity, is increasing. Depressive disorders represent a significant public health concern due to their prevalence, associated impairment, and economic impact [3]. Functional impairment in patients with MDD is very common; more than 90% of patients report functional impairment in at least one area during a major depressive episode, and this impairment can persist even after depressive symptoms improve. Incomplete functional recovery has been associated with an increased risk of relapse or recurrence of depression [4]. The use of pharmaceutical cognitive enhancers (PCEs) by people with MDD, in an effort to improve their memory, motivation, and concentration, may lead to drug misuse if used incorrectly [5]. 

The Diagnostic and Statistical Manual of Mental Disorders, Fifth Edition (DSM-5), has indicated that MDD is characterized by a marked change in mood and/or anhedonia, and the presence of several other psychophysiological changes, such as disturbed sleep, changes in appetite, fatigue, and a diminished ability to think or concentrate. Cognitive impairments are estimated to affect approximately two-thirds of individuals with MDD. The DSM-5 characterizes cognitive impairment as difficulty with thinking, concentrating, or making decisions [6].

Major depression is highly prevalent and in many of those affected, it takes a recurrent course with an increased probability of relapse with each episode: 50% after one, rising to between 70% and 90% following two and three episodes respectively [7]. Depression results from a complex interaction of social, psychological, and biological factors. There are effective treatments for moderate and severe depression, including medications and psychosocial interventions. One of the most commonly used psychosocial interventions is a mindfulness-based intervention [1].

### 1.2. Mindfulness-Based Cognitive Therapy and Major Depressive Disorder 

Research on mindfulness is constantly expanding, and findings suggest numerous positive outcomes. The practice of mindfulness has grown and developed over the past two decades. Mindfulness derives from its initial roots in Buddhism, and has traveled from the East to the West, where Jon Kabat Zinn applied its philosophies to behavioral interventions for medical problems. Mindfulness-based cognitive therapy (MBCT) is a group therapy involving 10 to 15 participants for six to eight weeks, created specifically to help manage stress and depressive symptoms. People can use the attitude of “non-judgment” from MBCT to avoid increasing rumination from negative thoughts. Mindfulness emphasizes direct experience, both mental and physical, rather than using cognition to elaborate the origin, meaning, or results from certain thoughts or feelings. The training of mindfulness enables the individual to move their attention from repeated negative thoughts to current personal experience in order to have a broader point of view, in addition to limiting rumination that might lead to relapse of depression [8,9]. 

MBCT is specifically designed to address cognitive vulnerability processes in those currently in remission from depression, but who are at high risk of relapse due to their previous history of recurrent depression [10] and is recommended for relapse prevention by the UK National Institute for Health and Clinical Excellence (NICE) [11]. The benefits of MBCT can be associated with improvement in cognitive control, regulation of emotions, positive mood, and acceptance, as it aims to teach patients to relate their experiences in order to improve unpleasant thoughts, feelings, and bodily sensations [12]. 

MBCT has not only been proven to treat the acute symptoms of depression [13], but also decreases recurrence of MDD significantly [8]. Research indicates that MBCT is effective in preventing relapse for up to 60 weeks when compared with both maintenance antidepressants and treatment as usual [14]. In contrast to other psychotherapies, MBCT relies on intensive mental training, starting with focused, sustained attention practices whilst building up to practices that encourage monitoring of all experiences, be they positive, negative, or neutral, with increased openness and acceptance [10]. Additionally, comprehensive depression treatments should include interventions that reduce suicide risk. It is still unclear if MBCT can reduce suicide risk compared to conventional treatment. However, through mindfulness training, MBCT may indirectly benefit individuals with recurrent suicidal ideation and behavior. Currently, the efficacy of evidence-based psychosocial interventions for reducing depression and suicides is limited [15].

Due to the inconsistencies in improvement of clinical outcomes from MBCT such as suicidal ideation, relapse rate, and overall depressive symptoms, it is important to synthesize current literature to better understand the state of science regarding the effects of MBCT in people with MDD. Therefore, it is important to identify the consistency of research methods between these studies using meta-analysis. 

### 1.3. Study Aim

The purpose of this systematic review and meta-analysis was to synthesize the effect of MBCT on depression and suicidal ideation among patients with MDD.

## 2. Methods

### 2.1. Identification of Relevant Literature

To obtain a more comprehensive review of the literature, we searched studies published in both Chinese and English from seven databases (Airiti Library, PsycINFO, CINAHL, Cochrane Library, PubMed/MEDLINE, ProQuest and the Index of the Taiwan Periodical Literature System) from the earliest available date to 31 December 2021. The search terms used were “mindfulness-based cognitive therapy (MBCT)” AND (“mindfulness-based” OR “mindfulness”) AND (“major depression” OR “major depressive disorder”) AND (“suicidal” OR “suicidal ideation”). Related articles identified in the database searches were also reviewed to identify additional pertinent studies. Only randomized controlled trials that examined MBCT for adults with major depressive disorder were included in the analysis. Duplicate publications, qualitative studies, and review articles were excluded. Figure 1 illustrates the search and selection process used in this study.

As a guideline for conducting the meta-analysis, we followed the Preferred Reporting Items for Systematic Reviews and Meta-Analyses statement (PRISMA) [12] to heighten the transparency and replicability of the findings.

### 2.2. Inclusion Criteria (Selection Criteria)

The following inclusion criteria were applied: (i) studies of individual randomized controlled trials (RCTs) of mindfulness-based cognitive therapy (MBCT) with a non-psychotherapeutic comparison condition (e.g., treatment as usual (TAU), clinical management, waitlist); (ii) adult participants diagnosed with a nonpsychotic, non-bipolar depressive disorder as principal diagnosis; and (iii) observer-rated depressive disorder diagnosed at baseline based on Research Diagnostic Criteria (RDC), DSM-IV/IV-TR/5 criteria, ICD-10 criteria, Chinese Classification of Mental Disorders (CCMD-3) criteria.

### 2.3. Search Outcome

A total of 1327 Chinese- and English-language publications were identified in the initial search. Primary appraisal of titles, abstracts, or full text was used to screen the articles. After a three-stage procedure involving screening of the titles, abstracts, and full text, 13 studies, all of which were RCTs, were included in the analysis (Figure 1 and Table 1).

### 2.4. Assessment of Study Quality

Two reviewers used the Cochrane risk of bias assessment tool (The Cochrane Collaboration, 2019) to assess literature quality independently, followed by joint checking to confirm the quality of individual studies. In cases of disagreement, study quality scores were determined following joint discussions with a third reviewer. The Cochrane risk of bias assessment tool was used to evaluate random sequence generation, allocation concealment, blinding of participants and personnel, blinding of outcome assessment, completeness of outcome data, selective reporting, and other sources of bias (Figure 2). Possible quality assessment results for each item are “low risk”, “high risk”, and “unclear”. The consistency of the two reviewers was assessed by calculating kappa values using IBM SPSS software (ver. 25.0; IBM Corporation, New York, NY, USA), with higher kappa values indicating greater consistency between reviewers.

Three reviewers used the Modified Jadad scale, which has shown excellent validity and reliability (infraclass correlation coefficient = 0.90), to appraise the quality of each clinical trial [16]. The scale has five items: randomization, appropriateness of randomization, blinding, appropriateness of blinding, and withdrawals-dropouts; total scores range from one to five, with a higher score indicating a higher quality study.

### 2.5. Quality Appraisal

#### 2.5.1. Data Extraction and Review Process

After completing literature searches, as outlined in Table 1, all hits (*n* = 1327) were saved in EndNote. Two reviewers independently extracted data. In cases of disagreement, the result was determined following discussions with a third reviewer. All potentially relevant articles were retrieved for full-text review (*n* = 27) which resulted in 13 studies included in the final review. Uncertainties regarding inclusion were checked again by the third team member, discussed, and resolved by consensus. A flow chart showing the process of study selection is given in Figure 1. 

#### 2.5.2. Data Abstraction and Synthesis

The study population and design, method-timing-duration of intervention and comparisons, main results, and quality of each study were abstracted and assessed (Table 1). Meta-analysis was used to pool data (e.g., mean, standard deviation, effect size, etc.) from the studies to determine the overall effect of the MBCT. Data were analyzed using Review Manager 5.3.5 software (The Cochrane Collaboration, Copenhagen, Denmark). Considering the difference in inclusion and exclusion criteria, variability in control or treatment intervention protocols, and variation in analysis (especially in handling withdrawals), we used a random effects model to estimate the overall effect conservatively, to avoid underestimating the variability of interventions [17]. To help readers evaluate the strength level of an intervention, we calculated the effect size of various intervention outcomes (e.g., depression, suicidal ideation, relapse to major depression, self-esteem, mindfulness skills, level of worry symptoms, rumination on sadness, medical coping mode, health survey, etc.) that cannot be pooled to compare using Review Manager 5.3.5 (Table 1).

### 2.6. Statistical Analysis

As we assumed heterogeneity between studies and in order to be able to make inferences beyond the observed sample of study populations, we used a random effects model for both research questions [18,19]. All statistical calculations were performed with Comprehensive Meta-Analysis 3 (CMA3). 

Heterogeneity was assessed by chi-square heterogeneity tests and *I*^2^ statistics. The chi-square value tests for statistically significant heterogeneity among trials; *p* values lower than 0.05 indicate heterogeneity. The *I*^2^ statistic is a measure for the amount of heterogeneity (range from 0 to 100%) that indicates what proportion of observed variability is due to real differences in effect sizes [18]. Higher *I*^2^ values indicate greater variability among trials than would be expected by chance alone. As a guideline, we used the *I*^2^ categorization provided by Higgins et al. (2003) [17], referring to *I*^2^ values of 25%, 50%, and 75% as low, moderate, and high (p. 559).

Publication bias was assessed by testing for funnel plot asymmetry [20]. The funnel plot is displayed in Figure 3, Figure 4 and Figure 5. Additionally, Rosenthal’s fail-safe number (Nfs) test was used to test for publication bias. Nfs = 19S-N (S = number of the studies with *p* < 0.05, N = number of the studies with *p* > 0.05). Tolerance level (TL) = 5K + 10 (K = total number of the studies). An Nfs value greater than the TL indicates minimal risk of bias, summary publication bias, and relative trustworthiness of the whole analysis [21]. Publication bias was only assessed for the second analysis, as the first analysis is a summary statistic of overall event rates.

## 3. Results

### 3.1. Characteristics of Included Studies

The literature search yielded 13 RCTs (published between 2008 and 2019) that met the selection criteria (Table 1). Overall, long-term follow-up data were available for 1159 patients. The number of depressed patients included in the follow-ups ranged from 28 [22] to 130 [23]. The majority of participants were female, ranging from 51.25% [24] to 79.0% [23], and the mean age ranged from 30.0 [25] to 48.77 years [26].

Twelve studies had implemented one or more non-psychotherapeutic comparison conditions such as a usual care group [13,22,23,24,25,26,27,28,29,30,31,32] or a clinical management for waitlist [33]. In terms of diagnostic criteria for MDD, six of the 13 studies had DSM-IV-TR criteria [13,19,22,27,28,33], one study utilized observer rated depressive disorder diagnosed at baseline based on Research Diagnostic DSM-IV criteria [23], one study used DSM-5 criteria [29], one study used ICD-10 criteria [24], three studies used Chinese Classification of Mental Disorders (CCMD-3) criteria [25,31,32], and one study used WHO criteria [30]. All studies followed-up patients who had experienced recurrent MDD but were currently remitted.

Follow-up times varied from 1.5 months [32] to 12 months [26] post-therapy. The number of intervention sessions ranged from six [32] to 12 [29]. Total intervention time was 16 h for six studies [13,22,23,25,26,28,33], 18 h for one study [28], 20 h for one study [23], 24 h for one study [29], 10–15 h for one study [32], 12–16 h for one study [31], and two studies did not mention the weekly intervention time [24,30]. The total home practice time after group therapy was 6 h for three studies [22,26,29], 3–6 h for one study [23], and seven studies did not mention the home practice time [13,24,25,26,27,28,30,31,32,33]. Only four studies recommended the number of people for participating in a group MBCT, ranging from 8 to 15 [23,24,29,31]. 

### 3.2. Study Quality

Of the 13 published studies, three (69.23%) were determined to be of good quality (Modified Jadad scale score > 5; Table 1). The quality of the 13 included studies is characterized in Figure 2 and Figure 3. The kappa value expressing consistency between the two reviewers was 0.933 (*p* < 0.0001), indicating a high level of agreement. The research team used the Modified Jadad score to evaluate study quality, with 27.07% of included studies having a score of 8.0 [22,26,27]; 27.07% scored 7.0 [23,28,33]; 27.07% scored 5.0 [13,24,31]; 7.69% scored 4.5 [29]; and 27.07% scored 1.0 [25,30,32]. Details regarding study quality are described in Table 1.

The risk of bias for each study is shown in Figure 2. Specifically, 77% (10/13) of the studies presented high risks of detection bias; 77% (10/13) presented high risks of detection bias and attrition bias; 38% (5/13) presented a high risk of performance bias; 15% (2/13) presented high risks of attrition bias, 7% (1/13) presented a high risk of bias in allocation concealment (selection bias); 7% (1/13) presented a high risk of reporting bias; and 7% (1/13) presented a high risk of bias in random-sequence generation (selection bias). Furthermore, 62% (8/13) of the studies had no other significant potential source of bias and 38% (5/13) showed heterogeneity of one or two basic attributes, which may have caused bias in the estimation of intervention effectiveness. Because 11 of 13 (85%) articles lacked selective reporting problems, the risk of reporting bias was low. In general, the overall quality of the 13 studies included in this paper was relatively high. Although this review included only randomized trials, 12 of the trials [7,22,23,24,25,26,27,28,30,31,32,33] contained sufficient methodological details for low risk of selection bias, while one trial [22] did not specify the inclusion and exclusion criteria, and was therefore considered to have a high risk of selection bias. Seven trials [7,22,23,26,27,28,33] described the use of random sequences for grouping, and so were at low risk for covert grouping, while the remaining 10 trials [7,24,25,26,28,29,30,31,32,33] did not report the detailed methods for randomization and were considered to be at high risk for hidden grouping. Seven trials [7,22,23,26,27,28,33] described the use of envelopes to achieve blinding and so were at low risk of performance bias, while the authors of the remaining five trials did not describe blinding in depth. Based on method descriptions, all studies were deemed to have low risk of detection bias. Three trials [23,30,33] reported patient drop-outs during the study period and so were considered to be at low risk of attrition bias, while drop-out rate was unclear in the remaining trials, which were therefore deemed to be at high risk of attrition bias. All trials provided sufficient details on patient characteristics to be considered at low risk of reporting bias.

### 3.3. Effectiveness of MBCT Using Meta-Analysis

#### 3.3.1. Depression

Although 13 RCTs were identified for the systematic review, four studies did not include baseline data or report means and standard deviations [13,28,30,33]. Therefore, nine RCTs were included in the meta-analysis.

Depression was measured by two scales, including the Beck Depression Inventory-II (BDI-II) Questionnaire in five studies and the Hamilton Rating Scale for Depression (HAMD) Questionnaire in six studies. Because different measures were used across studies, the effect of the MBCT could not be calculated together. The effect was reported based on each measure.

Depression as measured by the BDI-II Questionnaire showed a significant decrease after Mindfulness-Based Cognitive Therapy (MBCT) intervention in five studies [13,22,26,27,28]. The results of heterogeneity were examined by using Comprehensive Meta-Analysis (CMA), an analysis software package with an easy operation interface and enough power to meet the research needs. Meta-analysis is a statistical procedure that combines data from multiple studies. When a treatment effect (or effect size) is consistent across studies, a meta-analysis can be used to identify currently prevalent effects and changes in effect from one study to the next. Meta-analysis can be used to determine the reasons for the changes. The pooled standardized mean difference (SMD) between groups was 0.349 (95%CI = 0.073 to 0.626), and this overall effect was statistically significant (Z = 2.478, *p* = 0.013). Since variations in depression are complex, the heterogeneity between studies was significant (Q = 5.287, *p* = 0.259, *I*^2^ = 24.346), as determined by the random-effects model. The combined results indicate that the MBCT group had lower depression than the control group. The forest plot is displayed in Figure 6.

Depression as measured by the HAMD Questionnaire showed a significant decrease after MBCT intervention in six studies [23,24,25,29,31,32]. The pooled SMD between groups was 0.838 (95% CI: 0.044 to 0.426), and this overall effect was statistically significant (Z = 3.782, *p* = 0.000). Since variations in depression are complex, the heterogeneity between studies was significant (Q = 29.654, *p* = 0.000, *I*^2^ = 83.139), as determined by the random-effects model. The combined results indicate that the MBCT group had lower depression than the control group. The forest plot is displayed in Figure 7.

#### 3.3.2. Suicidal Ideation

Only three studies included suicidal ideation as an outcome variable. Suicidal ideation was measured using the Beck Scale for Suicide ideation (BSS) showing a significant decrease after MBCT intervention in these three studies [22,25,31]. The pooled SMD between groups was 0.910 (95% CI: 0.145 to 0.163), and this overall effect was statistically significant (Z = 2.387, *p* = 0.017). Since variations in suicidal ideation are complex, the heterogeneity between studies was significant (Q = 12.230, *p* = 0.002, *I*^2^ = 83.647), as determined by the random-effects model. The combined results indicate that the MBCT group had lower suicidal ideation than the control group. The forest plot is displayed in Figure 8.

## 4. Discussion

The purpose of this systematic review and meta-analysis was to evaluate the evidence regarding the efficacy of MBCT on depression and suicidal ideation among patients with MDD. Using the Modified Jadad scale appraisal checklist, most studies reviewed were rated between 5 and 8; hence, the quality of included studies was relatively high. The evidence from this systematic review supports the beneficial effect of MBCT on depression and suicidal ideation among patients with MDD. Statistically significant differences were found in 11 studies using the BDI-II [13,22,26,27,28] and the HAMD Questionnaire [23,24,25,29,31,32]. 

Rumination, a repetitive, uncontrolled, and negatively balanced cognitive process, is central to the maintenance of depression. MBCT teaches people to be more aware of their thoughts and feelings, and to accept them as mental activities rather than as right or wrong statements (i.e., it is non-judgmental). Such meditation practice improves the relationship between thoughts and emotions by shifting attention from repetitive thinking to focusing on the present moment, providing a broader perspective, thereby detaching affective processes from negative thinking [8,33]. With regular practice, MBCT can reduce ruminative thinking, which in turn reduces the risk of depressive relapse [33]. 

We found a significant heterogeneity between included studies using continuous outcome measures. It is possible that the use of different assessment scales contributes to the high heterogeneity. Another possible explanation for the high heterogeneity in our review is that included studies utilized different research designs, sample selection methods, intervention types, intervention times, practice frequency and evaluation methods. Although all studies used MBCT as the intervention, there were differences in inclusion and exclusion criteria, variations in choosing control or treatment interventions, and variations in analytical strategies (especially in handling withdrawals) among them. Therefore, the random effect model was used in the integrated analysis in an effort to avoid underestimating the variability of treatments [17].

Other reasons for the high inter-study heterogeneity in the studies included in this review may be related to variations in frequency of mindfulness practice. Our findings revealed that regular practices had an effect on depression scores. Regular practice of MBCT increases the thickness of the cortex in the areas of the somatosensory system, which is positively correlated with increased awareness of the body [34] A significant improvement in suicidal ideation was found in three studies using Relapse Signature of Suicidality Interview (ReSSI) scores [22,25,31]. Suicidal ideation is a very common and disturbing problem in MDD patients [1]. Suicidal ideation can negatively impact patients’ quality of life and trigger depression [1]. Another potential reason for high inter-studies heterogeneity may be sex differences; women with MDD are more prone to anxiety disorder and suicide ideation [35]. Studies indicates that the scales used by Chinese research institutions are all QSA self-assessment scales, while Western countries use different assessment scales. Differences in the scales used to assess suicidal ideation contributed to the high heterogeneity [15], this inference is consistent with previous research. In view of the above, we suggest that the assessment scale should be used consistently in the future.

MBCT usually combines cognitive therapy with intensive training in meditation [33]. The frequency of MBCT was one time per week in all studies in this review [13,22,23,24,25,26,27,28,29,30,31,32,33]. Total weekly intervention time was 16 hours for six studies [7,22,25,26,27,33], 18 h for one study [28], 20 h for one study [23], 24 h for one study [29], 10~15 h for one study [32], and 12~16 h for one study [31]. Two studies did not mention the weekly intervention time [24,30]. The total weekly home practice time after group therapy was 6 hours for three studies [22,26,29], 3~6 h for one study [23], and seven studies did not mention the home practice time [13,24,25,26,27,28,30,31,32,33]. It seems that a certain period of time at minimum is required for the benefits of MBCT to emerge. Through mindfulness training, MBCT may indirectly benefit individuals with recurrent suicidal ideation and behavior [15]. This inference is consistent with previous research. Based on the above, we suggest that more relevant research can be carried out in the future.

### 4.1. Implications

The intervention of MBCT to treat MDD is helpful in clinical practice. Based on this meta-analysis, we suggest that MBCT sessions lasting 1.5–2.5 h, five times per week for 8 weeks can effectively relieve depression and suicidal ideation indicators of MDD patients. These findings suggest that MBCT is a relatively convenient and effective way of preventing and improving block rumination in people with MDD. Furthermore, MBCT must be integrated as a treatment modality in MDD care to ensure continuity of care. Future research should test the evidence impact of using MBCT in order to further develop the field.

### 4.2. Limitations

Several limitations should be considered. First, despite a comprehensive literature search, only a limited number of studies met the inclusion criteria, which constrained the statistical power of this meta-analysis. Statistical analyses, such as meta-regression analysis and sensitivity analysis, could not be performed owing to the small sample size. Moreover, we could not perform subgroup analyses owing to the variation in research designs on MBCT. Furthermore, the included studies measured only the effect of MBCT intervention before and after comparison with a long-term follow up period. Therefore, the long-term effect of MBCT on depression and suicidal ideation among patients with MDD is unclear. Finally, this review study only includes studies published in English and Chinese, which may have resulted in the omission of relevant studies published in other languages.

## 5. Conclusions

The meta-effects of MBCT show significant improvements in depression and suicidal ideation among patients with MDD. These findings suggest that providing MBCT is feasible and effective for MDD patients. Future research should evaluate the long-term effects of MBCT.

## Figures and Tables

**Figure 1 ijerph-20-01555-f001:**
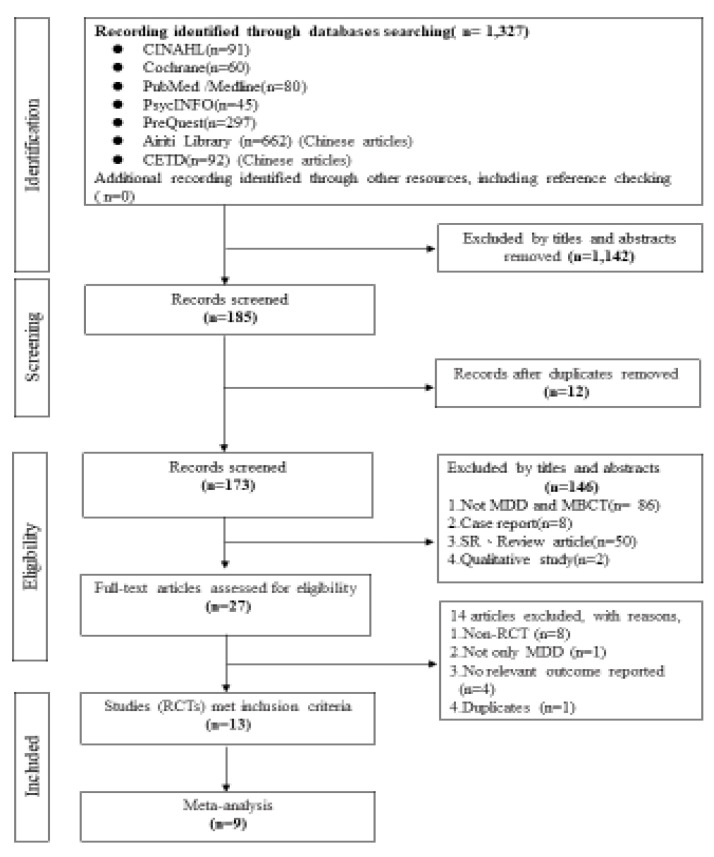
PRISMA diagram of study selection.

**Figure 2 ijerph-20-01555-f002:**
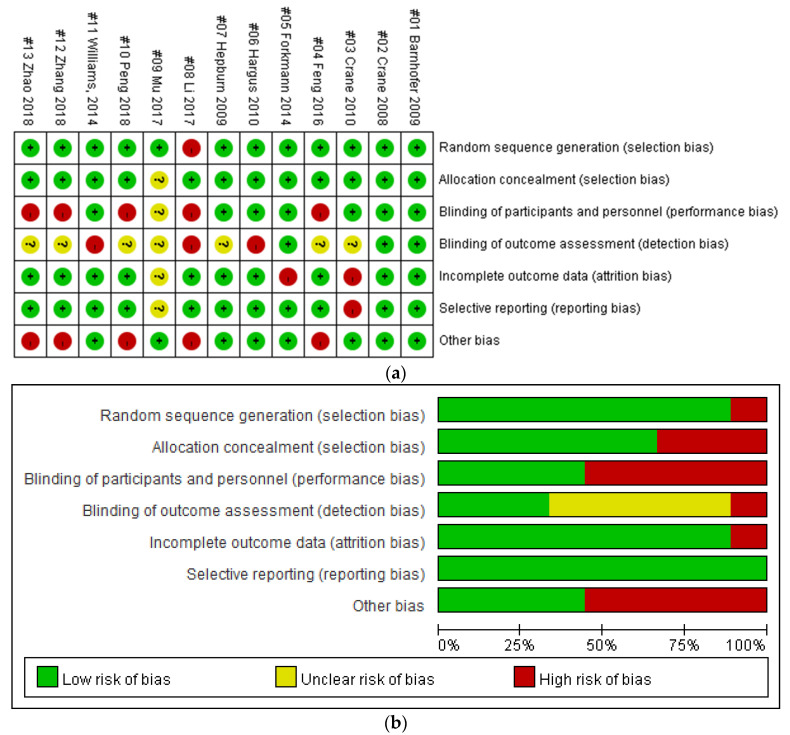
Risk of bias assessment: review authors’ judgments about each risk of bias item, presented as percentage of included studies. (**a**) Risk of bias summary. (**b**) Risk of bias graph (color figure can be viewed at wileyonlinelibrary.com accessed in 1 December 2022).

**Figure 3 ijerph-20-01555-f003:**
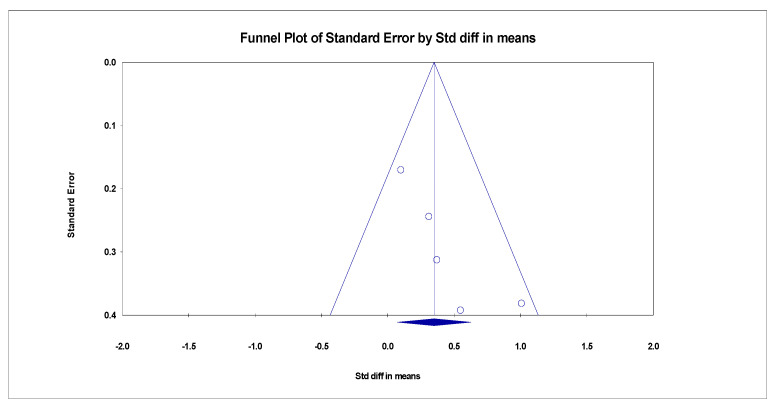
Funnel plot of BDI-II effect of MBCT on depression.

**Figure 4 ijerph-20-01555-f004:**
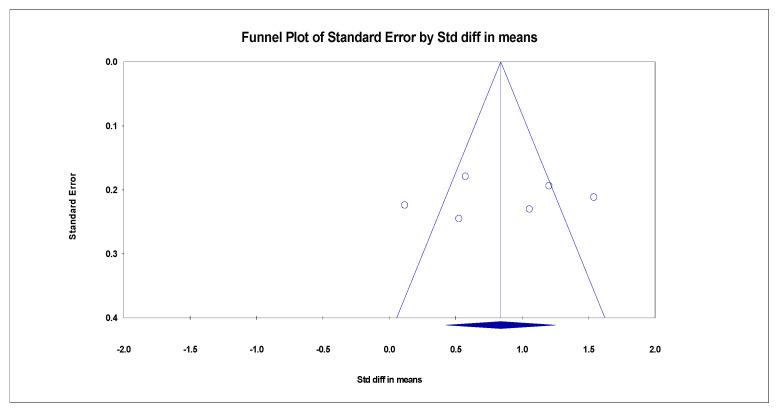
Funnel plot of HAMD effect of MBCT on depression.

**Figure 5 ijerph-20-01555-f005:**
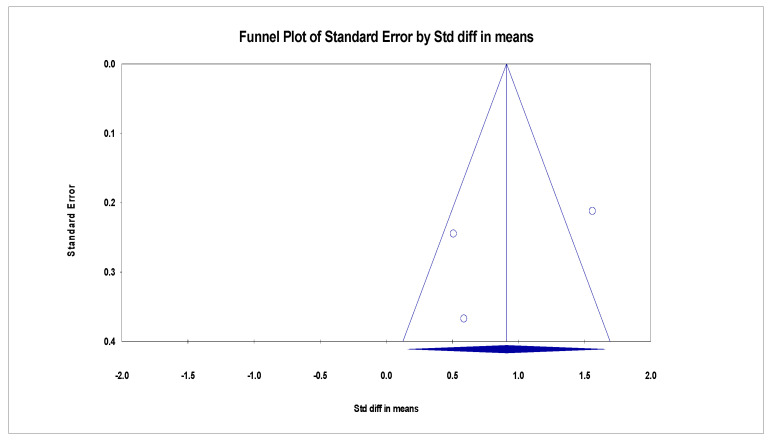
Funnel plot of suicidal ideation of MBCT on depression.

**Figure 6 ijerph-20-01555-f006:**
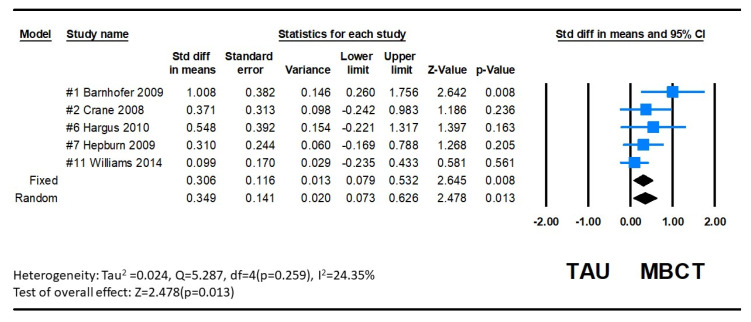
Forest plot of BDI-II effect of MBCT on depression.

**Figure 7 ijerph-20-01555-f007:**
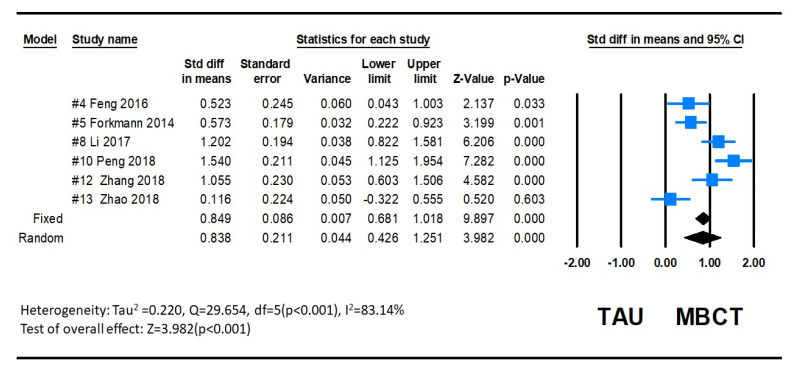
Forest plot of HAMD effect of MBCT on depression.

**Figure 8 ijerph-20-01555-f008:**
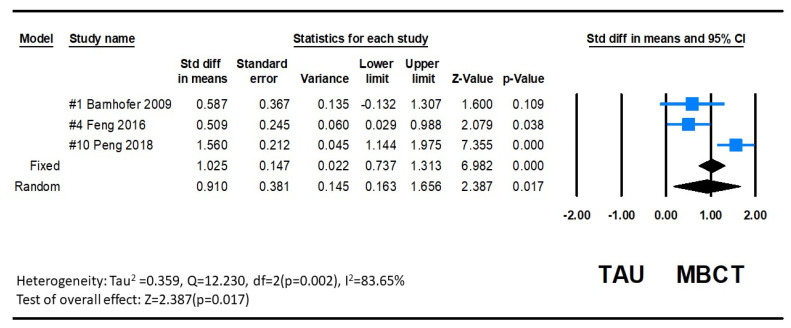
Forest plot of suicidal ideation effect of MBCT on depression.

**Table 1 ijerph-20-01555-t001:** Characteristics of included studies.

First Author, Year/Country	Diagnosis/ Assessed with	Sample Size; Attrition Rate (%); Gender (% Female)	Treatment Conditions/Age (Mean ± SD)	Treatment concept/Duration (Weeks)/Number of Group Participants	Treatment Classes (Hour/Day/Week)/Homework (Hour/Week)	Comparison Condition	Instruments/ Follow-up Duration	Main Result (Mean ± SD; *p* Value)	Modified Jadad Scale
#1 Barnhofer, 2009/England	Currently remitted recurrent MDD (DSM-IV-TR)/ history of at least 3 episodes of major depression/SCID	31; 28 analyzed completers; 67.86%	(1) MBCT+TAU (*n* = 16/14)/42.07 ± 11.34 (2) TAU (*n* = 15/14)/41.79 ± 9.52	MBCT/8 wks/NA	2/1/wk/6/wk	Naturalistic care, i.e., standard treatment	BDI-II BSS/post-treatment	1. Pre-assessments I 29.36 ± 9.66 vs. C: 31.32 ± 10.79/post-assessments I: 17.62 ± 10.94 vs. C: 28.86 ± 12.97; *p* = 0.03. 2. Pre-assessments E 2.21 ± 2.45 vs. C: 2.78 ± 2.08/ post-assessments I: 1.14 ± 1.79 vs. C: 2.42 ± 2.53.	8/8
#2 Crane, 2008/USA	Currently remitted recurrent MDD (DSM-IV-TR)/ history of at least 3 episodes of major depression	68; 68 analyzed completers; 67.86%	(1) MBCT (*n* = 33)/49.75 ± 8.11 (2) waitlist (*n* = 35)/40.44 ± 9.09	MBCT/8 wks/NA	2/1/wk/NA	Naturalistic care, i.e., standard treatment	BDI-II /post-treatment	1. Pre-assessments I: 16.58 ± 14.23 vs. C: 12.78 ± 9.83/post-assessments I: 8.40 ± 12.59 vs. C: 12.90 ± 11.76/, *p* = 0.004.	8/8
#3 Crane, 2010/USA	Currently remitted recurrent MDD (DSM-IV-TR)/ history of at least 3 episodes of major depression	91; 35 analyzed completers; NA	(1) MBCT (*n* = 35)/49.75 ± 8.11 (2) waitlist (*n* = 36)/40.44 ± 9.09	MBCT/8 wks/NA	2.5/1/wk/NA	Naturalistic care, i.e., standard treatment	BDI-II /post-treatment	1. Pre-assessments I: 12.85 ± 10.89, *p* > 0.05.	7/8
#4 Feng, 2016/China	Currently remitted recurrent MDD	69; 69 analyzed completers; 55.07%	(1) MBCT+TAU (*n* = 34) (2) TAU (*n* = 35)	MBCT/8 wks/NA	2/1/wk/NA	Naturalistic care, i.e., standard treatment	SDS HAMD /post-treatment	1. Pre-assessments I50.92 ± 11.29 vs. C: 51.48 ± 10.94/post-assessments I: 40.65 ± 9.17 vs. C: 45.59 ± 10.21; *p* < 0.05 2. Pre-assessments I: 26.56 ± 10.45 vs. C: 27.07 ± 1019/post-assessments I: 8.83 ± 6.14 vs. C: 12.37 ± 7.32; *p* < 0.05.	1/8
#5 Forkmann, 2014/USA	Currently remitted recurrent MDD (DSM-IV)/SCID/HDRS > 7	130; 130 analyzed completers; MBCT79.0%/TAU73.0%	(1) MBCT+TAU (*n* = 64)/44.6 ± 9.7 (2) TAU (*n* = 66)/43.2 ± 9.5	MBCT/8 wks/10–15	2.5/1/wk/3–6/wk (in the weekly group sessions, participants received CDs with guided exercises and were assigned daily)	Naturalistic care, i.e., standard treatment	HDRS/ post-treatment	1. Pre-assessments I: 10.27 ± 3.69 vs. C: 10.21 ± 3.55/ post-assessments I: 7.14 ± 4.81 vs. C: 9.68 ± 4.04; *p* = 0.00.	7/8
#6 Hargus, 2010/USA	Currently remitted recurrent MDD (DSM-IV-TR)	90; 68 analyzed completers; MBCT75%/TAU62.5%	(1) MBCT (*n* = 33)/42.07 ± 11.04 (2) TAU (*n* = 35)/41.69 ± 9.90	MBCT/9 wks/NA	2/1/wk	Naturalistic care, i.e., standard treatment	BDI-II /post-treatment, 3, months	1. I: 30.35 ± 9.93 vs. C: 32.37 ± 11.18; *p* > 0.05.	7/8
#7 Hepburn, 2009/USA	Currently remitted recurrent MDD (DSM-IV-TR)	125; 68 analyzed completers;; 73.53%	(1) MBCT+TAU (*n* = 33)/48.77 ± 9.04 (2) TAU (*n* = 35)/41.24 ± 9.00	MBCT/8 wks	2/1/wk; 360/wk	Naturalistic care, i.e., standard treatment	HAMD-17 /post-treatment, 3, 6, 9, 12 months	1. Pre-assessments I: 15.62 ± 13.84 vs. C: 12.83 ± 9.59/post-assessments I: 8.67 ± 12.00 vs. C: 12.25 ± 11.14, *p* < 0.01	8/8
#8 Li, 2017/China	Currently remitted recurrent MDD (DSM-5)/ HAMD > 7	126; 126 analyzed completers, 53.96%	(1) MBCT+TAU (*n* = 61)/39.95 ± 12.50 (2) TAU (*n* = 65)/41.05 ± 15.41	MBCT/8 wks/10–12	2/1/wk/6/wk (total 12 weeks)	Naturalistic care, i.e., standard treatment	HAMD-17 /post-treatment	1. Pre-assessments I: 17.39 ± 5.70 vs. C: 18.10 ± 5.57/post-assessments I: 5.83 ± 4.74 vs. C: 11.19 ± 4.18; *p* < 0.01.	4.5/8
#9 Mu, 2017/China	WHO criteria of major depression	80; 45 analyzed completers; 55%	(1) MBCT+TAU (*n* = 40)/37.69 ± 7.06 (2) TAU (*n* = 40)/36.41 ± 8.62) y	MBCT/8 wks	NA/NA		HAMD-17	1. Post-assessments; *p* < 0.005	1/8
#10 Peng, 2018/China	Currently remitted recurrent MDD (CCMD-3))/ HAMD > 17	116; 116 analyzed completers; 51.72%	(1) MBCT+TAU (*n* = 58)/37.86 ± 5.12 (2) TAU (*n* = 58)/38.12 ± 5.24	NA	1.5–2/1/wk/NA	Naturalistic care, i.e., standard treatment	HAMD-17 /post-treatment	1. Pre-assessments I: 25.86 ± 3.85 vs. C: 26.54 ± 4.12/post-assessments I: 19.45 ± 3.67 vs. C: 14.12 ± 3.24; *p* < 0.05.	5/8
#11 Williams, 2014/USA	Currently remitted recurrent MDD (DSM-IV-TR)	257; 257 analyzed completers; NA	(1) MBCT+TAU (*n* = 99)/NA (2) CPE+TAU (*n* = 103)/NA (3) TAU (*n* = 55)/NA	MBCT/8 wks/NA	2/1/wk/NA	Naturalistic care, i.e., standard treatment	BDI-II HAMD-17 SCID /post-treatment, 3, 6, 9, 12 months	1. Measure of depressive symptomatology: BDI-II: no significance 2. Residual depressive symptomatology: HAMD: no significance 3. Suicidality questions of the SCID: no significance	5/8
#12 Zhang, 2018/China	Currently remitted recurrent MDD (CCMD-3))	86; 86 analyzed completers; 56.97%	(1) MBCT+TAU (*n* = 43)/35.74 ± 3.98 (2) TAU (*n* = 43)/4.92 ± 4.33	MBCT/6 wks	0.33–0.5/5/wk/NA	Naturalistic care, i.e., standard treatment	HAMD /post-treatment	1. Pre-assessments I: 28.11 ± 8.84 vs. C: 27.65 ± 8.69/ post-assessments I: 11.45 ± 5.71 vs. C: 17.29 ± 5.36; *p* < 0.05.	1/8
#13 Zhao, 2018/China	Currently remitted recurrent MDD (ICD-10)/HAMD > 17	80; 80 analyzed completers; 51.25%	(1) MBCT+TAU (*n* = 41)/42.93 ± 1.77 (2) TAU (*n* = 39)/43.45 ± 1.82	MBCT/8 wks/8	NA/NA	Naturalistic care, i.e., standard treatment	HAMD-17 /post-treatment	1. Pre-assessments I: 26.16 ± 5.19 vs. C: 26.22 ± 5.08/post-assessments I: 14.01 ± 1.70 vs. C: 14.22 ± 1.91; *p* < 0.05.	5/8

Note: MDD: major depressive disorder; E: experimental group; C: control group; W: waitlist group; MBCT = mindfulness-based cognitive therapy; CPE = cognitive psychological education; TAU = treatment as usual; BDI-II: Beck Depression Inventory-II; HAMD: Hamilton Rating Scale for Depression; SCID: Hamilton Rating Scale for Depression; BSS: Beck Scale for Suicide Ideation; DDS: Suicidal Ideation: Self-Rating Form of the Dutch Version of the Inventory of Depressive Symptoms; DSM-IV-TR: Diagnostic and Statistical Manual of Mental Disorders (DSM-IV-TR); CCMD-3: Chinese Classification of Mental Disorders.

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
