# Peer review of "Effects of Mindfulness-Based Cognitive Therapy on Major Depressive Disorder with Multiple Episodes: A Systematic Review and Meta-Analysis"

_ijerph, 2023, doi:10.3390/ijerph20021555_

Round 1

Reviewer 1 Report (Previous Reviewer 1)

I agree with all the revisions made. I suggest to accept the paper in present form.

This manuscript is a resubmission of an earlier submission. The following is a list of the peer review reports and author responses from that submission.

Round 1

Reviewer 1 Report

The paper contains sufficiently new and suitable information, and it adheres to the journal’s standards. The topic and level of formality are appropriate for the journal`s readership. Its style and readability are suitable. There is a huge amount of information given throughout the article, but I would suggest revising the paper. 

The methodological concept is clear, and the selected methodology is scientifically appropriate. This is a valuable approach.

I miss recent relevant literature in this area. The literature review is very poor. I suggest citing The role of different behavioral and psychosocial factors in the context of pharmaceutical cognitive enhancers’ misuse. Healthcare DOI: 10.3390/healthcare10060972 to add the pharmaceutical cognitive enhancers’ misuse, which could be a valuable approach to this topic.

Figure 2 should be better explained.

The results are analyzed appropriately, but they should be presented more clearly.  

Further, I recommend rewriting the conclusions. The concluding remarks should be more specific and better explained. I suggest adding more future directions.

In summary, the article is sufficiently interesting to warrant publication, but it needs major revision. Please follow all the comments above.

Author Response

1.a. The paper contains sufficiently new and suitable information, and it adheres to the journal’s standards. The topic and level of formality are appropriate for the journal`s readership. Its style and readability are suitable. There is a huge amount of information given throughout the article, but I would suggest revising the paper. .

Response: Thank you for your encouragement and support.

1.b. The methodological concept is clear, and the selected methodology is scientifically appropriate. This is a valuable approach.

Response: Thank you for your encouragement and support.

1.c. I miss recent relevant literature in this area. The literature review is very poor. I suggest citing The role of different behavioral and psychosocial factors in the context of pharmaceutical cognitive enhancers’ misuse. Healthcare DOI: 10.3390/healthcare10060972 to add the pharmaceutical cognitive enhancers’ misuse, which could be a valuable approach to this topic.

Response: Thank you very much for this valuable suggestion. We have added a citation (Tomažicˇ, T&C elofiga, A.K., 2022) and justification for using approach at the beginning of the introduction section on Page 2.

1.d. Figure 2 should be better explained.

Response: Thank you for this suggestion. We have made this revision at the results section on Page 11 and 12 

1.e. The results are analyzed appropriately, but they should be presented more clearly. I recommend rewriting the conclusions. The concluding remarks should be more specific and better explained. I suggest adding more future directions.

Response: We appreciate these comments. We have made this revision at the discussion section. Please see Page 14. 

1.f. In summary, the article is sufficiently interesting to warrant publication, but it needs major revision. Please follow all the comments above.

Response: Thank you for your encouragement and support. We appreciate these comments. We will do our best to follow all the comments above to revision.

Reviewer 2 Report

Introduction:

Justification:

In the paragraphs where he presents Mindfulness-Based Cognitive Therapy he presents studies that seem to support the efficacy of this therapy, without going into limitations of the same that reveal biases or deficiencies that invite us to think about other results.

Given the assertion of the presence of inconsistencies, it would be appropriate for you to show evidence of these inconsistencies.

Please could you support this assertion with studies?

It is suggested that you justify with evidence the gaps in knowledge that you intend to clarify.

As a final sentence in the introduction, you add "Also, current review of the effect of MBCT on suicidal ideation, relapse rate, and depression has not been fully covered in English databases."

In this case there is a published review that you can consult at the following link: https://www.sciencedirect.com/science/article/pii/S0165032722010977

I think it is interesting to take it into account for its justification and to review the articles included in it.

Methodology

It is suggested that you indicate whether the review has been registered and where the review protocol can be accessed.

The use of the 2009 PRISMA Statement has become obsolete. Authors are suggested to update their paper according to what is described in the current statement which can be found at the following link: https://www.bmj.com/content/372/bmj.n71

Regarding the flowchart, it is not very clear why 158 items are removed from the process with a gap between 185 and 27, could you please explain. Thank you in advance.

Results

It is suggested to review the "main result" column of table 1. Use the same format to present the results as described in the column header. Please check the font size in the table.

It is suggested that you revise the confidence interval of the standardized mean difference for the relationship between MBCT and depression measured with the BDI-II.

It is suggested that you provide the chi-square and Nfs results indicated in section 2.6 for the results.

It is indicated in the text that the asymmetry was assessed with the funnel plot but it is not provided. It would be interesting to assess it for possible publication bias.

Discussion

The objective proposed at the beginning of the study and the objective described in the discussion section differ, it is suggested that both have the same purpose. 

Given the limitations of the study, it is recommended that the statements and recommendations made in the discussion section be moderated. Some of the limitations rightly highlighted by the authors could lead to a type I error.

On the other hand, the study has related the intervention to depression and ideation scores, not to rumination directly, so it is suggested that this be taken into account in the implications section.

Author Response

2.a. Introduction

- In the paragraphs where he presents Mindfulness-Based Cognitive Therapy he presents studies that seem to support the efficacy of this therapy, without going into limitations of the same that reveal biases or deficiencies that invite us to think about other results.

Given the assertion of the presence of inconsistencies, it would be appropriate for you to show evidence of these inconsistencies.

Please could you support this assertion with studies?

It is suggested that you justify with evidence the gaps in knowledge that you intend to clarify.

As a final sentence in the introduction, you add "Also, current review of the effect of MBCT on suicidal ideation, relapse rate, and depression has not been fully covered in English databases."

In this case there is a published review that you can consult at the following link: https://www.sciencedirect.com/science/article/pii/S0165032722010977

I think it is interesting to take it into account for its justification and to review the articles included in it.

Response: Thank you for the comments. We have added a citation (Zhang et al., 2022) and justification for using approach at the end of the Introduction on Page 2 and 3.

2.b. Methodology

- It is suggested that you indicate whether the review has been registered and where the review protocol can be accessed

Response: Thank you for this suggestion. We have applied for Prospero through www.crd.York.ac.uk with the registration number.

2.c. Methodology

-The use of the 2009 PRISMA Statement has become obsolete. Authors are suggested to update their paper according to what is described in the current statement which can be found at the following link: https://www.bmj.com/content/372/bmj.n71

Regarding the flowchart, it is not very clear why 158 items are removed from the process with a gap between 185 and 27, could you please explain. Thank you in advance.

Response: Thank you for these suggestions. We have revised the process accordingly. Please see Figure 1 on Page 4.  158 items were removed from the process because we removed duplicate recorders (n=12) and excluded titles or abstracts that did not meet the search criteria (n=146). We also corrected this Figure 1 on Page 4.

2.d. Results

-It is suggested to review the "main result" column of table 1. Use the same format to present the results as described in the column header. Please check the font size in the table

Response: Thank you for this suggestion. We have revised the sentence accordingly (Table 1) on Page 5-7. 

2.e. Results

-It is suggested that you revise the confidence interval of the standardized mean difference for the relationship between MBCT and depression measured with the BDI-II

Response: Thank you for this suggestion. Because the intervention methods considered in the study are all MBCT, but the intervention time of each study is different, and the confidence intervals in studies 2, 6, and 7 are relatively large, so it is more appropriate to use random model for analysis. We also conduct sensitivity analysis. This s approach had no effect on the results, and the p-value still reached a statistically significant difference.

2.f. Results

- It is suggested that you provide the chi-square and Nfs results indicated in section 2.6 for the results. It is indicated in the text that the asymmetry was assessed with the funnel plot but it is not provided. It would be interesting to assess it for possible publication bias.

Response: Thank you for this recommendation. We have added the Figure 3、Figure 4& Figure 5 on Page10 

2.g. Discussion

-The objective proposed at the beginning of the study and the objective described in the discussion section differ, it is suggested that both have the same purpose. 

Given the limitations of the study, it is recommended that the statements and recommendations made in the discussion section be moderated. Some of the limitations rightly highlighted by the authors could lead to a type I error. On the other hand, the study has related the intervention to depression and ideation scores, not to rumination directly, so it is suggested that this be taken into account in the implications section.

Response: Thank you for these great questions. It has been revised regarding the objective proposed at the beginning of the study and the objective described in the discussion section differ. Please see the discussion on Page 13.

In order to avoid the type I error, we corrected the term significant improvement in depression to statistically significant differences on Page 13.

Reviewer 3 Report

Thanks for the opportunity to review this intersting paper. This study synthesizes the effect of mindfulness-based cognitive therapy (MBCT) on depression and suicidal ideation among patients with major depressive disorder. The meta-effects of MBCT among patients with MDD showed significant improvement in depression and suicidal ideation. However, I have several major concerns for this study. 

1. Introdcution: This part is too lenthy, and did not sufficiently illustrate the neccesity and originality of this study. Why you studyMBCT? in fact there are very strong evidence exist on the effects of  MBCT on MDD, what's your study going to add?

2.Methods: This study did not have sensitivity analyis to validate the results.

3. The outline of the discussion is weird. quality of the studies is part of results, not discussion. In fact, in discussion, you should highlight the findings, compare your findings, explain your findings. The authors neglect the comparison with previous studies.

Author Response

3.a.  Introduction

- This part is too lengthy, and did not sufficiently illustrate the necessity and originality of this study. Why you study MBCT? in fact there are very strong evidence exist on the effects of MBCT on MDD, what's your study going to add?

Response: Thank you for these great questions. We have made this revision in the Introduction section on Page 2 and 3. 

3.b. Method

-This study did not have sensitivity analysis to validate the results.

Response: Thank you for this suggestion. According to Higgins, Thompson, Deeks, and Altman (2003)*, it is suggested that no sensitivity analysis is required, since the number of studies mentioned in the article is less than 30. So it is not written in the article. However, we conduct sensitivity analysis per suggestions. It indicated that regardless of deletion any included study there is no effect on the results, and the p-value still reaches a statistically significant difference.

*Higgins, J.P.T., Thompson, S. G., Deeks, J.J., & Altman, D. G. (2003). Inconsistency in meta-analyses. British Medical Journal, 327(7414), 557-560.https://doi.org/10.1136/bmj.327.7414.557

3.c. The outline of the discussion is weird. quality of the studies is part of results, not discussion. In fact, in discussion, you should highlight the findings, compare your findings, explain your findings. The authors neglect the comparison with previous studies.

Response: Thank you very much for these great questions. We have made this revision at the discussion section. Please see Page13- 14. 
